# ALITA: GENERALIST AGENT ENABLING SCALABLE AGENTIC REASONING WITH MINIMAL PREDEFINITION AND MAXIMAL SELF-EVOLUTION

## ABSTRACT

Recent advances in large language models (LLMs) have enabled agents to autonomously perform complex, open-ended tasks. However, many existing frameworks depend heavily on manually predefined tools and workflows, which hinder their adaptability, scalability, and generalization across domains. In this work, we introduce **Alita**—a generalist agent designed with the principle of *"Simplicity is the ultimate sophistication,"* enabling scalable agentic reasoning through *minimal predefinition* and *maximal self-evolution*. For *minimal predefinition*, Alita is equipped with only one component for direct problem-solving, making it much simpler and neater than previous approaches that relied heavily on hand-crafted, elaborate tools and workflows. This clean design enhances its potential to generalize to challenging questions, without being limited by tools. For *Maximal self-evolution*, we enable the creativity of Alita by providing a suite of general-purpose components to autonomously construct, refine, and reuse external capabilities by generating task-related model context protocols (MCPs) from open source, which contributes to scalable agentic reasoning. Notably, Alita achieves 75.15% pass@1 and 87.27% pass@3 accuracy, which is top-ranking among general-purpose agents, on the GAIA benchmark validation dataset, 74.00% and 52.00% pass@1, respectively, on Mathvista and PathVQA, outperforming many agent systems with far greater complexity. Our code is open-sourced.

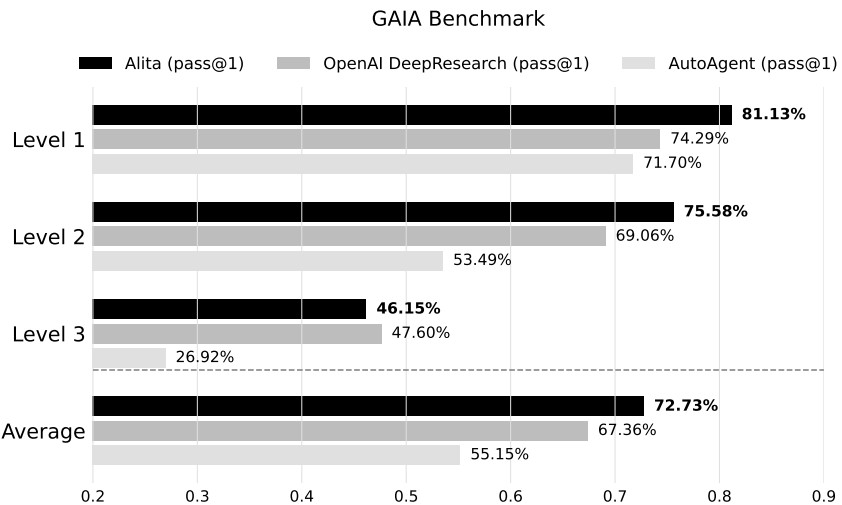

Figure 1: Performance of Alita, OpenAI DeepResearch[1], and AutoAgent[2].

# 1 INTRODUCTION

> *"Simplicity is the ultimate sophistication."*                   — Leonardo da Vinci

Large language models (LLMs) have rapidly evolved from merely generating text to autonomous agents capable of independently planning and executing complex tasks on behalf of users with limited human oversight [3]. These capabilities have enabled a wide range of applications, ranging from travel planning [4], computer use [5; 6; 7], to the multi-step research tasks [8]. To support such diverse and demanding tasks, a new class of systems called generalist agents has emerged. These agents are designed to handle a wide range of domains and tasks through a unified architecture, allowing them to generalize beyond task-specific solutions, such as OpenAI Deep Research [1] and Manus.

However, most of the current general-purpose agents heavily rely on large-scale manual engineering, including tediously designed workflows, considerable pre-defined tools, and hardcoded components [9; 10]. This reliance introduces several critical limitations: i) It is impractical, if not impossible, to predefine all the tools required for the wide variety of real-world tasks an agent might encounter (*incomplete coverage*); ii) Many complex tasks require agents to creatively compose new tools or leverage existing ones in novel ways while pre-designed workflow and hardcoded components constrain this compositional flexibility and inhibit the development of adaptive behaviors (*limited creativity and flexibility*); iii) It is not always the case that the interface or environment of different tools are compatible with the agent (*mismatch*). For example, many useful tools are not written in Python, which makes it difficult, though not entirely impossible, for them to be pre-connected to the mainstream agent frameworks that are primarily written in Python. Together, these challenges ultimately hinder the scalability, adaptability, and generalization of existing generalist agents.

In contrast to the prevailing trend of growing complexity, we propose a radically simple design philosophy built on two principles: i) *Minimal Predefinition*: Equip the agent with only a minimal set of core capabilities, avoiding manually engineered components for specific tasks or modalities; ii) *Maximal Self-Evolution*: Empower the agent to autonomously create, refine, and reuse external capabilities as needed. We instantiate this vision through Alita, a generalist agent built with a single core capability (i.e., the web agent) and a small set of general-purpose modules that enable self-directed capability expansion. Specifically, we take advantage of the Model Context Protocols (MCPs) [1] which is an open protocol that standardizes how different systems provide context to LLMs, and empower Alita to dynamically generate, adapt, and reuse MCPs based on the demands of each task rather than relying on static, predefined tools. This shift from manually designed capabilities to on-the-fly MCP construction unlocks a new path for building agents that are simple yet profoundly capable.

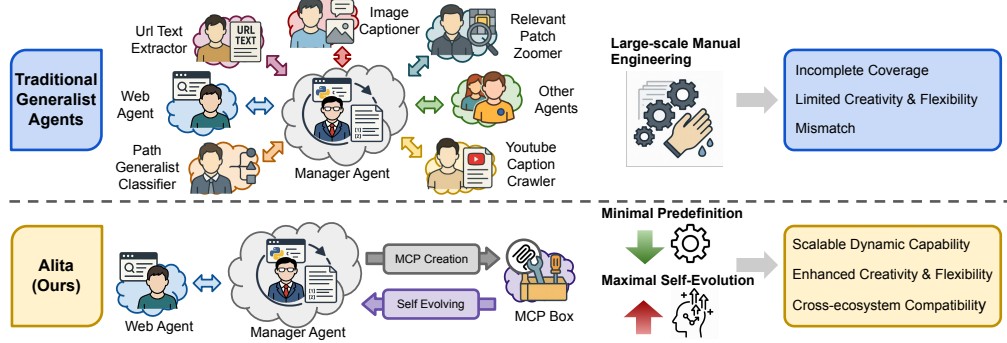

Figure 2: Comparison between Traditional Generalist Agents and Alita. Traditional generalist agents heavily rely on large-scale manual engineering while Alita adheres to minimal predefinition and maximal self-evolution.

We conduct comprehensive experiments on several benchmarks to assess Alita in real-world applications, especially on the most popular GAIA [11]. Alita proves that simplicity is not a constraint, but a

---

[1]https://www.anthropic.com/news/model-context-protocol

strength, and that creative agent behavior can emerge from a design that prioritizes autonomy over manual engineering. To sum up, our key contributions can be summarized as follows.

- We propose a new agent architecture centered on minimal predefinition and maximal self-evolution, challenging conventional design norms in generalist agents, aiming to call for a more scalable and generalizable agent framework.
- We present Alita, a generalist agent that achieves scalable agentic reasoning with a radically simple design.
- We empirically demonstrate that Alita, despite using no complex predefined tools and workflows for specific tasks, outperforms many systems with significantly more handcrafted complexity on the GAIA benchmark. We achieve 75.15% pass@1 and 87.27% pass@3, surpassing OpenAI's Deep Research with 67.36% pass@1 and ranking top among all general-purpose agents.

## 2 RELATED WORKS

### 2.1 GENERALIST AGENT

The concept of a Generalist Agent aims to construct an AI agent system capable of collaboratively completing a variety of complex tasks in a real-world environment. OWL [9] introduces a method that decomposes complex tasks into subtasks and dynamically allocates them to worker nodes with specialized toolkits. Omne [12] proposes a multi-agent collaborative development framework, where each agent possesses an independent system structure, enabling autonomous learning and the storage of a comprehensive world model to build an independent understanding of the environment. OpenAI Deep Research[2] employs reinforcement learning for training on real-world tasks, aiming to provide precise and reliable research reports for knowledge-intensive tasks. A-World [13] offers a highly configurable, modular, and scalable simulation environment, allowing developers to flexibly define and integrate different types of AI agents. The Magentic-One [14] framework merges the Magentic and Autogen systems, distinguishing between the micro-level LLM-driven function generation and the macro-level multi-agent orchestration, resulting in a clearer and more efficient construction of agent systems. Alita, also a Generalist Agent, allows for the minimal use of predefined tools and workflows for direct problem-solving, yet still achieves impressive performance across diverse tasks.

### 2.2 AUTO GENERATING AGENT

Auto-generating agents aim to enhance the versatility of agents by enabling them to autonomously generate tools, agents, or workflows tailored to specific tasks. AutoAgents [15], for instance, generate multiple agents, each playing a distinct role, to handle the corresponding subtasks. OpenHands [16] offers an event-driven architecture that allows agents to interact with the environment like human developers, thereby enabling the creation of custom workflows. AFlow [17] redefines workflow optimization as a search problem, and the optimal workflow is identified and executed through a search process. AutoAgent [2], as an autonomous agent operating system, permits agents to manage system-level operations and file data autonomously. In Alita, agents are empowered to automatically generate diverse, specialized, and highly accurate MCPs to aid in the completion of specific tasks, while also providing resources for future executions.

### 2.3 TOOL CREATION

Tool Creation enables agents to autonomously create tools to assist in task execution, either on their own or with external support. CRAFT [18] utilizes GPT-4 to generate a set of code snippets that function as tools, which are then retrieved and used in the system. TroVE [19] maintains a collection of high-level functions, which are automatically generated, extended, and periodically pruned to optimize program generation. CREATOR [20] decouples the abstraction of tool creation from the actual execution, allowing LLMs to address tasks at different levels of granularity. AutoAgent [2] enables agents to autonomously create new tools based on task requirements, incorporating information gathered through web searches and integrating these tools into their workflow. OpenHands [16]

---

[2]https://openai.com/index/introducing-deep-research/

allows agents to create code scripts in a human-like manner during interaction with the environment to assist in task completion. In comparison, Alita enables MCP creation, which provides additional benefits, including better reusability and easy environment management over tool creation.

## 2.4 MCP

The Model Context Protocol (MCP) is a standard proposed by Anthropic, designed to unify the connection between AI systems and external data sources and services. RAG-MCP [21] enhances the efficiency and accuracy of agents by retrieving the most relevant tools from a large collection, based on the task description, within the database composed of MCP descriptions. After tool generation, Alita wraps the generated valid tools into MCPs for subsequent use, facilitating reuse by itself and other agents.

## 3 METHODS

We propose Alita, a generalist agent enabling scalable agentic reasoning with minimal predefinition and maximal self-evolution to tackle diverse and complex tasks. Figure 3 illustrates the framework of Alita. In contrast to generalist agents that typically depend on extensive manually-designed tools and workflows [9; 10], the manager agent in Alita solely orchestrates the web agent using only basic tools. Through this approach, our framework enables Alita to plan task-specific tools through brainstorming. It then utilizes a Web Agent to search for helpful open-source libraries and other resources related to these tools. Leveraging the search results, Alita autonomously generates new tools and configures the necessary environments to enhance its capabilities and effectively solve tasks. During this process, if any issues arise with the newly generated tools or their environments, Alita can provide feedback and self-correct, improving the quality of the generated tools. Furthermore, the new tools can be encapsulated as MCP servers for future reuse. With the aid of MCPs, Alita can generate increasingly powerful, diverse, and complex MCPs, thus establishing a self-reinforcing cycle. Therefore, Alita autonomously expands its capabilities through continuous MCP integration.

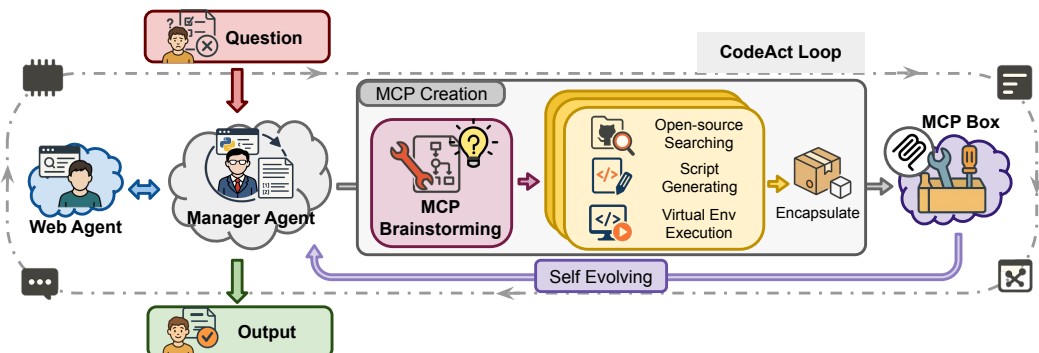

Figure 3: The architecture of Alita. Upon receiving a question, the Manager Agent initiates an iterative CodeReAct loop to analyze tasks, identify functional gaps, and trigger MCP Brainstorming for creative synthesis. The system dynamically performs open-source searching, script generation, and virtual environment execution to construct task-related functions. Useful ones are encapsulated into reusable MCPs and stored in the MCP Box. Throughout this process, the Manager Agent collaborates with the Web Agent for external information retrieval and continuously integrates intermediate results until a final output is produced. This process enables Alita to self-evolve without relying on a huge hand-crafted, elaborate tools and workflows.

### 3.1 EXECUTION PIPELINE

Each task commences with the construction of an augmented prompt that incorporates the original query. The manager agent (Sec. 3.2) then initiates a multi-step reasoning process to address the task at hand. Throughout this process, the agent may query external sources via the web agent (Sec. 3.3), plan and synthesize new tools (Sec. 3.4), and execute them within isolated environments (Sec. 3.4.4).

Upon successful tool generation and accurate result formulation, the corresponding script is transformed into a MCP and stored in the internal tool registry for future reuse. All reasoning steps, intermediate code, and final outputs are systematically logged to facilitate comprehensive analysis.

## 3.2 MANAGER AGENT

The Manager Agent functions as the central coordinator within framework of Alita. When receiving a task prompt, the manager agent initially calls the `MCP Brainstorming` to determine whether additional tools are needed and which specific tools are required. Then the manager agent decomposes the task into subtasks and dispatches them to web agent or generates the required external tools to complete the subtask. When necessary, the manager agent utilizes the information retrieved by the web agent to generate the required new tools along with their corresponding environment configuration instructions. After collecting all intermediate results, the manager performs final aggregation and response formulation.

**Tool Usage**. In contrast to traditional systems that rely on extensive predefined toolkits, the manager agent embraces Alita's minimal philosophy by employing concise but powerful toolkits, including `MCP Brainstorming`, `ScriptGeneratingTool` and `CodeRunningTool`. Specifically, `MCP Brainstorming` detects functional gaps, identifies necessary supplementary tools and outlines tool specifications; `ScriptGeneratingTool` obtains tool specification outlines and then generates appropriate tools tailored to the task requirements; `CodeRunningTool` executes generated code in isolated environments and caches the output for potential MCP servers generation. Additionally, the manager uses `TextInspectorTool` to inspect textual content, `BaseAnswerTool` to format answers, and `Visualizer` to interpret visual files such as charts or images. These tools are intelligently invoked in response to the task's evolving demands, ensuring adaptive and efficient problem-solving.

## 3.3 WEB AGENT

The web agent retrieves relevant information from external sources when internal knowledge is insufficient. It is particularly effective for tasks requiring the retrieval of domain-specific code or documentation. With a lightweight, text-based web interface and modular navigation tools, the web agent traverses multiple websites, extract relevant segments, and return reasonable URLs or raw content.

**Tool Usage**. The agent utilizes `SimpleTextBrowser` as its web interface and page-level control tools: `VisitTool`, `PageUpTool`, and `PageDownTool` to navigate webpages. For query-based lookups, it applies `GoogleSearchTool` for open web search and `GithubSearchTool` to identify reusable open-source tools. This design supports real-time code retrieval and context-aware tool planning.

## 3.4 MCP CREATION COMPONENT

To enable the creativity of the agent, we design three tools collaboratively contributing to the MCP creation process.

### 3.4.1 MCP BRAINSTORMING

Since LLMs often exhibit overconfidence in their capabilities [22], we introduce `MCP Brainstorming` to conduct preliminary capability assessment by providing both the task and the description of current framework. We designed specialized prompts to facilitate accurate self-assessment of the agent's capabilities. Moreover, when `MCP Brainstorming` identifies insufficient capabilities of the framework to complete the task, it provides references for tool generation to bridge the capability gap. This provides prior guidance for subsequent tool selection and task planning required to accomplish given objectives.

### 3.4.2 SCRIPTGENERATINGTOOL

The `ScriptGeneratingTool` is a code-building utility designed for constructing external tools. It receives explicit subtask descriptions and suggestions for code construction from the manager agent, and potentially useful GitHub links obtained via the web agent, which can provide information such as `README.md` files or code snippets from GitHub to guide the script generation process. Furthermore,

`ScriptGeneratingTool` generates the environment script to create the required environment for the code running and the cleaning script to clean up redundant files and environments generated after script execution. Therefore, `ScriptGeneratingTool` ensures that the generated scripts are valid, self-contained, and executable, making them suitable for deployment in the given task, and reusable in the future.

### 3.4.3 CODERUNNINGTOOL

The `CodeRunningTool` validates the functionality of the generated script by executing it within an isolated environment. If the execution produces the expected results, the tool is registered in the system as a reusable MCP. This process also supports iterative refinement, allowing for error inspection and subsequent code regeneration to improve the script's performance.

### 3.4.4 ENVIRONMENT MANAGEMENT

Upon retrieving or generating a candidate tool, the system activates the environment planner module. This module parses the relevant repository or script metadata such as `README.md`, `requirements.txt`, and shell scripts using the `TextInspectorTool`. It extracts and validates the dependencies and setup instructions to construct an isolated execution profile. Subsequently, a new Conda environment is created with a unique name (typically derived from the task ID or a hash of the repository path), and dependencies are installed using `conda install` or `pip install`.

All runtime environments are initialized locally in parallel, obviating the need for administrative privileges or containerization technologies. This approach ensures high compatibility across various tasks while preserving the portability of the system. During execution, the environment is explicitly activated prior to invoking the code interpreter, thus ensuring both isolation and reproducibility.

In the event of a failure during environment initialization—due to issues such as missing packages, syntax errors in setup scripts, or unavailable dependencies—Alita activates an automated recovery procedure. This procedure attempts various fallback strategies, including relaxing version constraints or identifying the minimal set of dependencies required for functionality. If these recovery attempts are unsuccessful, the tool is discarded, and the failure is logged for offline analysis and future investigation. This enables Alita to self-correct its designed tools, thereby generating more accurate and robust solutions.

## 4 EXPERIMENTS

### 4.1 EXPERIMENT SETTING

### 4.1.1 BENCHMARKS

To evaluate the general task-handling capabilities of Alita, we conducted extensive testing across multiple agent benchmarks.

**GAIA [11]:** GAIA is a benchmark designed to assess the capabilities of general-purpose AI assistants. It consists of 466 real-world scenario-based questions covering daily tasks, scientific reasoning, web browsing, and tool usage. While these tasks are conceptually simple for humans, they are challenging for most advanced AI systems.

**Mathvista [23]:** MathVista is a comprehensive benchmark designed to evaluate the mathematical reasoning capabilities of foundation models within visual contexts. It can effectively evaluate the model's capabilities in visual comprehension, mathematical reasoning, programming, and other related skills. Due to limitations in resources, we randomly selected 100 samples from the dataset.

**Pathvqa [24]:** PathVQA is a medical visual question answering dataset. It can effectively assess the agent's capabilities across multiple dimensions, including visual understanding, spatial Reasoning, medical Knowledge search or integration, and natural language processing. Due to limitations in resources, we randomly selected 100 samples from the dataset.

| Agent | GAIA | | | | Mathvista | PathVQA |
|---|---|---|---|---|---|---|
| | level1 | level2 | level3 | total | | |
| *Alita (%)* | | | | | | |
| pass@1 | 81.13 | 75.58 | 46.15 | 72.73 | **74** | **52** |
| pass@2 | 88.68 | 80.23 | 53.85 | 78.79 | - | - |
| pass@3 | 96.23 | 86.04 | 65.38 | **86.06** | - | - |
| *Baselines (%)* | | | | | | |
| Octotools | - | - | - | 18.40 | 68 | 47 |
| ODR-Smolagents | 67.92 | 53.49 | 34.62 | 55.15 | 65 | 42 |
| AutoAgent | 71.70 | 53.49 | 26.92 | 55.15 | - | - |
| OWL | 84.91 | 67.44 | 42.31 | 69.09 | - | - |
| A-World | 86.79 | 69.77 | 34.62 | 69.70 | - | - |
| OpenAI-DR | 74.29 | 69.06 | 47.60 | 67.36 | - | - |

Table 1: Performance comparison of Alita and baseline agent systems on the GAIA, Mathvista, and PathVQA benchmarks. ODR-Smolagents refers to the Open Deep Research agent in the Smolagents framework. OpenAI-DR refers to the OpenAI's Deep Research agent. The table presents the accuracy at different levels of difficulty for GAIA, as well as the overall performance on Mathvista and PathVQA. The pass@1, pass@2, and pass@3 denote the accuracy achieved by running the Alita framework 1, 2, and 3 times, respectively, and selecting the best answer. Alita outperforms all baseline agents across the GAIA levels, achieving the highest total accuracy.

### 4.1.2 BASELINES

We include a variety of baselines for comparison. For the GAIA benchmark, there are more baselines available on the GAIA leaderboard[3].

**Octotools [10]:** OctoTools is a recent framework designed to streamline multi-tool workflows in complex computational tasks. With over 10 standardized tool cards encapsulating various functionalities, the agent gains powerful capabilities to handle multi-domain tasks.

**Open Deep Research of Smolagents[4] [25]:** Open Deep Research is an open-source agent developed under Hugging Face's Smolagents project, designed to automate complex multi-step research tasks. By combining LLMs with an autonomous execution framework, it enables web browsing, information extraction, file processing, and data computation.

**AutoAgent [2]:** AutoAgent framework is a zero-code platform designed to facilitate the creation, customization, and deployment of agents powered by LLMs. By providing a natural language interface, it allows users to develop multi-agent systems, design workflows, and integrate tools without requiring technical expertise.

**OWL [9]:** OWL is an open-source, multi-agent framework built on the CAMEL-AI platform, designed to support the automation of complex real-world tasks through dynamic agent collaboration. OWL decomposes tasks into specialized sub-tasks, each of which is managed by a distinct agent type—such as UserAgents, AssistantAgents, and ToolAgents.

**A-World [13]:** A-World is an open-source multi-agent system framework designed to simplify the construction, evaluation, and deployment of general multi-agent tasks. Through its modular design, the framework supports autonomous decision-making, tool usage, and collaboration among agents.

**OpenAI Deep Research[5]:** OpenAI's Deep Research is an advanced AI agent integrated with ChatGPT, designed to autonomously perform multi-step research tasks by synthesizing information from diverse online sources. This agentic framework excels in generating comprehensive reports on complex topics and has shown superior performance on benchmarks.

---

[3]https://huggingface.co/spaces/gaia-benchmark/leaderboard

[4]https://huggingface.co/blog/open-deep-research

[5]https://openai.com/index/introducing-deep-research/

| Model Configuration | Level 1 | Level 2 | Level 3 | Total |
|---|---|---|---|---|
| ODR-smolagents + GPT-4o(No Alita MCPs) | 33.96% | 29.07% | 11.54% | 27.88% |
| ODR-smolagents + GPT-4o(With Alita MCPs) | 39.62% | 36.05% | 15.38% | **33.94%** |

Table 2: Comparison of performance between ODR-smolagents with and without Alita-generated MCPs. The results are reported at different GAIA levels: Level 1, Level 2, Level 3, and the average. Each column corresponds to the performance at the respective GAIA levels. The reuse of Alita-generated MCPs can enhance the performance of other agents.

## 4.2 RESULTS

We run three rounds of testing on GAIA and achieved the best performance on the GAIA leaderboard, surpassing other agent systems. Alita achieves 72.73% pass@1 and 86.06% pass@3 accuracy, which ranks top 1 among all open-source frameworks temporarily, on the GAIA benchmark, outperforming many agent systems with far greater complexity. Alita also achieves 74.00% and 52.00% pass@1, respectively, on Mathvista and PathVQA, outperforming Octotools and Open Deep Research by smolagents. More detailed results are shown in Table 1.

## 5 ANALYSIS

### 5.1 REUSE OF ALITA-GENERATED MCPS

#### 5.1.1 OVERVIEW

We collect the MCPs generated from running the GAIA dataset using Alita in conjunction with powerful models (Claude-3.7-Sonnet and GPT-4o). The benefits of reusing Alita-generated MCPs are **two-fold**. First, these MCPs can be reused by other agent frameworks and improve their performance since Alita, instead of human developers, designs a set of useful MCPs fit to GAIA by trial and error. Second, these MCPs can be reused by agents with smaller LLMs and **significantly** improve the performance. The reuse of auto-generated MCPs for agents with smaller LLMs can be viewed as a new way of distillation from larger LLMs. Traditionally, distillation might be fine-tuning smaller LLMs on data generated by larger LLMs. In comparison, the reuse of MCPs generated from agents with larger LLMs is much easier, cheaper, and faster than traditional distillation.

#### 5.1.2 REUSE BY OPEN DEEP RESEARCH-SMOLAGENTS

We run open Deep Research-smolagents [25] on GAIA with and without Alita-generated MCPs based on GPT-4o. The results are presented in Table 2. From this experiment, we observe that the reuse of Alita-generated MCPs results in better performance compared to the base framework for all difficulty levels. This demonstrates that Alita can generate very useful MCPs, which can be provided to other agents, helping them enhance their capabilities and solve problems that would otherwise be unsolvable. Additionally, the consistent improvement across all difficulty levels indicates that Alita's MCPs provide generalizable utility rather than just addressing specific edge cases in the dataset.

#### 5.1.3 REUSE BY BASE AGENT ON SMALLER LLM

We reuse MCPs in the base framework, i.e., ODR-smolagents [25], without the MCP creation component in Alita, and also with some extra pre-defined tools used in ODR-smolagents based on GPT-4o-mini. The results are presented in Table 3.
From this experiment, we observe that the reuse of Alita-generated MCPs significantly improves performance over the base framework based on a smaller LLM. This is because the Alita-generated MCPs can be considered MCPs distilled from powerful models, which are made available for agents on smaller LLMs. This helps bridge the gap between the agents on smaller LLMs and agents on larger LLMs in certain domains, thereby enhancing its task-processing capabilities. Especially for Level 3, we observe a particularly dramatic improvement with the accuracy tripling from **3.85**% to **11.54**%. This substantial improvement on the most challenging problems demonstrates that Alita-generated

| Model Configuration | Level 1 | Level 2 | Level 3 | Average |
|---|---|---|---|---|
| Base Framework + GPT-4o-mini (No Alita MCP) | 32.08% | 20.93% | 3.85% | 21.82% |
| Base Framework + GPT-4o-mini (With Alita MCP) | 39.62% | 27.91% | 11.54% | **29.09%** |

Table 3: Comparison of performance between the base framework on GPT-4o-mini, with and without Alita-generated MCPs. The results are reported at different GAIA levels: Level 1, Level 2, Level 3, and the average. Each column corresponds to the performance at the respective GAIA levels. The reuse of Alita-generated MCPs significantly enhances the performance of agents on smaller LLMs.

| Model Configuration | Level 1 | Level 2 | Level 3 | Total |
|---|---|---|---|---|
| Alita (Claude-3.7-Sonnet, GPT-4o) | 81.13% | 75.58% | 46.15% | **72.73%** |
| Alita (GPT-4o-mini) | 54.72% | 44.19% | 19.23% | 43.64% |

Table 4: Comparison of performance between Alita(Claude-3.7-Sonnet,GPT-4o) and Alita(GPT-4o-mini). The results are reported at different GAIA levels: Level 1, Level 2, Level 3, and the average. Each column corresponds to the performance at the respective GAIA levels. The integration of a smaller model significantly reduces the performance.

MCPs are especially valuable for complex reasoning tasks where agents on smaller LLMs typically struggle the most. The MCPs effectively encapsulate sophisticated problem-solving capabilities that the smaller model can leverage without needing to develop the full reasoning chain independently.

## 5.2 ALITA ON SMALLER LLM

We hypothesize that **Alita will be even stronger with the increasing coding and reasoning capabilities of LLMs in the future**. To validate our performance, we run Alita on GAIA using GPT-4o-mini instead of Claude-3.7-Sonnet. The results can be found in Table 4. Different to the experiment in Section 5.1.3, the agent doesn't have distilled MCPs - the agent on GPT-4o-mini model must generate its own MCPs. The results are presented in Table 4.

From this experiment, on the one hand, we observe that Alita, after replacing the models with GPT-4o-mini, performs significantly worse on GAIA. This substantial performance gap highlights the critical role of the underlying models' coding capabilities. On the other hand, the performance of Alita increases rapidly as the capabilities of the underlying models improve. We can expect that with future updates to the LLMs, Alita's performance will continue to strengthen, surpassing its current capabilities. The design of future generalist agents might be much simpler in the future, without any predefined tools and workflows for direct problem-solving. Instead, human developers might focus on designing modules for enabling and stimulating the creativity and evolution of generalist agents.

## 5.3 CASE STUDY

To investigate Alita's workflow when tackling tasks, we conducted a case study on its approach to solving a Level 3 difficult problem in GAIA. The details of this process are presented in Appendix B. From the case study, we observe that Alita is able to perform a structured MCP brainstorming session based on the task at hand, effectively identifying and utilizing relevant resources to implement a feasible MCP that aids in completing the task.

## 6 CONCLUSION

In this work, we introduced **Alita**, a generalist agent designed with the principles of minimal predefinition and maximal self-evolution. By significantly reducing reliance on manually predefined tools and workflows for direct solving, Alita leverages creative, autonomous capabilities in real time, facilitating scalable agentic reasoning. Our approach demonstrates that simplicity in design does not undermine, but rather enhances, the performance and adaptability of generalist agents.

## 7 REPRODUCIBILITY STATEMENT

We will open-source our code and evaluation scripts upon publication. All datasets, model settings are described in the paper, enabling researchers to reproduce all reported experiments and results.

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

## A    THE USE OF LLMS

LLMs did not play an important role in this paper's research ideation or writing to the extent that they should be regarded as a contributor. In the experiments, LLMs are the main experimental object.

## B    DETAILED CASE STUDY

---

**Case Study**: YouTube 360 VR Video Subtitle Extraction

**Question ID:** 0512426f-4d28-49f0-be77-06d05daec096
**Question:** In the YouTube 360 VR video from March 2018 narrated by the voice actor of Lord of the Rings' Gollum, what number was mentioned by the narrator directly after dinosaurs were first shown in the video?
**Our Answer:** 100000000
**Correct Answer:** 100000000
**Is Correct:** Yes
**Generated MCP:** YouTube Video Subtitle Crawler

**Alita Workflow:**
**1. MCP Brainstorming:** Alita propose the development of a "YouTube Video Subtitle Crawler" MCP, which should automate the extraction of subtitles from a given YouTube video. This involves scraping the subtitles of the video and processing them to isolate the relevant text after the event in question.
**2. Web Agent Execution:** To implement the subtitle extraction, a search is conducted in open-source repositories to find relevant tools that can assist in extracting YouTube video transcripts. An appropriate tool, the youtube-transcript-api, is identified from the following GitHub repository:

        https://github.com/jdepoix/youtube-transcript-api

**3. Manager Agent:** The Manager Agent synthesizes the information from the GitHub repository and proceeds to write a Python function that leverages the youtube-transcript-api to retrieve the transcript of the video with corresponding environment setup instructions. The environment setup and installation steps are defined as follows:

```
conda create -n youtube_transcript
conda activate youtube_transcript
pip install youtube-transcript-api
```

The Python code to retrieve the video transcript is as follows:

```
from youtube_transcript_api import YouTubeTranscriptApi
# Initialize the API
ytt_api = YouTubeTranscriptApi()
# Retrieve the transcript
video_id = ...
transcript_list = ytt_api.list('video_id')
...
```

**4. Manager Agent Execution:** Leveraging the Python code and the established environment, the Manager Agent successfully packaged the YouTube Video Subtitle Crawler MCP. Subsequently, this MCP was employed to efficiently scrape the subtitles from the video, enabling the extraction of the relevant content. After analyzing the content, the correct number (100000000) mentioned by the narrator following the dinosaur scene is extracted from the transcript.
**5. Final Output:** The number "100000000" is identified as the correct answer.

---

## C    LIMITATIONS

Alita highly relies on the coding capability of LLM. When the LLM's coding capability is really poor, our method will perform worse than traditional generalist agent.

# D    EXPERIMENT SETTING UP

In Table 1, we selected three benchmarks, including GAIA (validation), Mathvista (test-mini), and PathVQA (validation), to evaluate the performance of Alita and other baseline agents.

For model configuration details, Alita's manager agent and MCP brainstorming use a large model based on Claude-3.7-sonnet, while the remaining agents and tools employ a large model based on GPT-4o. The baseline Smolagents also uses Claude-3.7-sonnet for its manager agent, with the rest of the agents and tools relying on GPT-4o. The baseline Octotool is built on Claude-3.7-sonnet.

Due to resource constraints, we tested pass@1, pass@2, and pass@3 performance on 165 questions from the GAIA validation set. For Mathvista test-mini and PathVQA validation, we randomly selected 100 questions each to evaluate the pass@1 performance of Alita, Octotools, and Smolagents. The results are all demonstrated in Table 1.

