# OpenReview forum: "Alita: Generalist Agent Enabling Scalable Agentic Reasoning with Minimal Predefinition and Maximal Self-Evolution"
_ICLR.cc/2026/Conference — ICLR 2026 Conference Desk Rejected Submission_

### Official Review · Reviewer_RaUK · 2025-10-31

**Soundness:** 3
**Presentation:** 3
**Contribution:** 3
**Rating:** 4
**Confidence:** 4

**Summary:**

This paper presents Alita, a generalist agent system built on the idea of "minimal predefinition + maximal self-evolution." Instead of packing the agent with tons of pre-built tools like most systems do, Alita starts with just a web browsing agent and creates tools on-the-fly using Model Context Protocols (MCPs). On the GAIA benchmark, it gets 72.73% pass@1 accuracy, beating OpenAI's Deep Research at 67.36%.

**Strengths:**

- Clear idea, small starting system. The “minimal preset + self-evolution” design cuts manual plumbing and avoids a big tool zoo.

- Good results. On GAIA, Alita beats many strong baselines; results on MathVista and PathVQA are also competitive.

- Transfer to other stacks. Reusing Alita’s MCP tools inside another agent gives clear gains, which suggests a nice path for sharing skills across systems.

**Weaknesses:**

- Scalability over time is untested. As the MCP store grows, retrieval speed, conflicts, and tool choice errors may rise. No long-run or high-load study.

- Ablations are missing. We don’t know how much each module (brainstorming, web search, env recovery, retries) actually helps.

- Key method details are thin. How MCPs are checked, stored, deduped, and retrieved is not clear enough to reproduce or scale.

- Safety and compliance. The agent searches the web, pulls code, and runs it. The paper does not detail sandbox rules, license checks, data contamination, or supply-chain guards.

-

**Questions:**

- What about cost? Pass@3 means 3x cost. If baselines had the same budget, would Alita still win?

- Do Alita-made MCP tools help on tasks beyond GAIA (e.g., other domains or data sources)?

---

> ### Author Response · Authors · 2025-12-03
>
> ## Response to W1
> The performance reported in the paper is the case that we don't store any MCPs but it already has very good performance.
>
> ## Response to W2
> In this framework, ablation study doesn't apply. Because any of the ablation will make other modules nonsense.
>
> ## Response to W3
> We haven't implement the MCP post-processing in this version. Just dynamically constructing the MCP according to the task description and executing it to solve the task. Other details can be seen in the code.
>
> ## Response to W4
> Thank you for pointing out this. Safety and compliance might be a big problem. We don't consider it when implementing this. Attack from evil tasks may bring Alita into trouble though we don't meet these problems on the current datasets.
>
> ## Response to Q1
> Baselines with 3x cost, Alita can still win. This is because baselines don't have the ability to generate MCPs(tools) for different tasks. Also, tool overload will be a big problem for them.
>
> ## Response to Q2
> Alita-made MCP tools generated on GAIA-validation can be helpful on GAIA-test. Also, some of the MCP tools such as video understanding tools are reused by other work such as HistAgent for historical reasoning to solve HistBench.

---

### Official Review · Reviewer_Bw6M · 2025-10-31

**Soundness:** 2
**Presentation:** 2
**Contribution:** 2
**Rating:** 2
**Confidence:** 3

**Summary:**

The paper presents a generalist agent framework that emphasizes two design principles: Minimal Predefinition (using only one core capability, the web agent, and a small set of general-purpose tools) and Maximal Self-Evolution (enabling the agent to autonomously construct, refine, and reuse external capabilities). Instead of relying on hardcoded tools or workflows, it generates and manages new tools by itself through “MCP brainstorming”. The generated MCPs can be cached for future reuse. The proposed method achieves superior results on GAIA, MathVista, and PathVQ datasets.

**Strengths:**

1. Simple design and flexibility. The framework only uses a small set of tools, avoiding labour-intensive tool definition. This also makes it more flexible in different domains.
2. The method outputs baselines on several benchmarks while being less complex. The authors also showed that the generated MCPs (tools) can be reused in other scenarios, such as other agent frameworks, smaller LLMs.

**Weaknesses:**

1. Although the framework does not rely on predefined tools, it still needs to generate task-specific tools and manage the resulting tool set. Therefore, the advantage of simplicity is only evident at the initial stage. In essence, compared with other agent frameworks, this approach merely adds a tool creation module. From this perspective, the contribution and novelty of the method are rather incremental.

2. The experiments are not sufficiently thorough. Although the proposed framework achieves strong results, there is no analysis explaining why the improvement occurs. For example, the key highlights of the framework are tool creation and reuse. It would be great to analyze whether the generated tools are actually better or more functional than the predefined ones across different tasks, and whether the model effectively reuses these generated tools. Such analyses should be included in the paper to substantiate the effectiveness of the proposed approach.

**Questions:**

See weaknesses

---

> ### Author Response · Authors · 2025-12-03
>
> ## Response to W1
> We strongly disagree on this point. The current performance in table 1 is zero-shot without saving and reusing the MCPs in the MCP box. For each task, Alita dynamically generate the related MCPs and execute them to solve the task. Then the pass@1 is reported. Also, pass@3 doesn't re-use any MCP in the MCP box.
>
> Alita beats a lot of startups or big teams where a lot of engineers who manually write a lot of tools to get good performance on GAIA. We adds a tool creation module and deletes a lot of unnecessary tools. A general agent should dynamically write tools instead of predefining a lot of tools to be general. Otherwise, it is only a domain-specific agent or a GAIA-specific agent.
>
> ## Response to W2
> We disagree. We already cover a lot of analysis in the paper. The reason for the improvment is very obvious. Alita can come up with more appropriate MCPs for different tasks than human engineers carefully dig into GAIA benchmarks and write pre-defined tools hoping to work on a batch of tasks.
>
> For example, one task in GAIA is to ask, "How many slides in this PowerPoint presentation mention crustaceans?" If the predefined PPT processing tool just converts all contents into text, it may fail to extract the page information and answer the question. However, Alita will dynamically create an appropriate PPT processing tool and wrap up as an MCP that is strong enough for solving this task.
>
> Another example is about the video analysis tool. Many tasks in GAIA are about YouTube video understanding. We observe that some general-purpose agent predefines the video analysis tool as a YouTube transcript crawler tool. However, some of the YouTube video understanding task requires a deeper understanding of the videos. Just reading from the transcript cannot solve the problem. Alita can create an MCP that reads the video frame by frame to solve some harder tasks for video understanding. This is task-specific MCP creation depending on the difficulty of the task. We are not experts in agent for video understanding, but Alita can show us how to construct the video understanding tool. This video understanding component is also reused by another work. We also find that even Gemini 3 cannot solve these harder video understanding task currently because in their training data for agent, the predefined tool is not implemented in the way of frame-by-frame video understanding.

---

### Official Review · Reviewer_WgV5 · 2025-11-03

**Soundness:** 3
**Presentation:** 3
**Contribution:** 4
**Rating:** 6
**Confidence:** 3

**Summary:**

This paper introduces Alita, a generalist agent framework designed to overcome the scalability and adaptability limitations of agents that rely on extensive, manually predefined tools. Alita is built on two principles: "Minimal Predefinition," where the agent starts with only a core set of components (like a web agent), and "Maximal Self-Evolution," where it autonomously creates, refines, and reuses new capabilities as needed.
Alita's workflow involves a Manager Agent that identifies a capability gap, brainstorms a new tool, and uses a Web Agent to find relevant open-source code. It then generates a script, tests it in a virtual environment, and encapsulates the new, successful tool as a "Model Context Protocol" (MCP) for storage and future reuse. The authors evaluate Alita on the GAIA, Mathvista, and PathVQA benchmarks, showing strong performance. They also demonstrate that the MCPs generated by Alita can be reused to improve the performance of other agents and smaller, less capable LLMs.

**Strengths:**

S1. The core design principle of "Minimal Predefinition and Maximal Self-Evolution" is an elegant and original contribution that addresses the reliance on manually-defined tools in agent development.

S2. Alita achieves nice performance, outperforming several baselines on the GAIA benchmark and also showing good results on Mathvista and PathVQA.

S3. The auto-generated MCPs are not just useful for Alita but can be exported to improve other agents and distill reasoning capabilities from large LLMs to smaller ones is a significant finding. This is especially true for the dramatic improvement seen on difficult (GAIA Level 3) tasks.

S4. The paper is well-written and easy to understand.

**Weaknesses:**

W1. The framework highly relies on the coding and reasoning abilities of top-tier LLMs (Claude-3.7-Sonnet and GPT-4o). The results in Table 4 show that when a smaller model (GPT-4o-mini) is used to generate MCPs, the performance drops drastically. This suggests the "minimal predefinition" approach is not yet practical without access to the powerful models (especially powerful coding models).

W2. The paper does not provide an analysis of the cost of MCP creation. How many tokens, how much wall-clock time, and how many self-correction attempts  are required, on average, to generate a new, functional MCP?

W3. While the average GAIA score is high, the bar chart in Figure 1 shows Alita (46.15%) performing slightly worse than OpenAI DeepResearch (47.60%) on Level 3 tasks. This point is not discussed, but it might suggest that for the most complex problems, manually-engineered tools still have an edge.

W4. There is a lack of newer baselines, such as the Tongyi Deep Research series (WebSailor, etc.).

**Questions:**

Q1. Could the authors clarify the discrepancy in the GAIA pass@1 score? The abstract states 75.15% , but Figure 1 and Table 1 show 72.73%. Which number is correct?

Q2. What is the computational and token cost of the "MCP Creation" loop? Could the authors provide data on the average number of attempts, tokens, or time taken to successfully generate and validate a new MCP for different tasks?

Q3. The workflow relies on the Web Agent finding useful open-source libraries. Does the timeliness of the searched documents affect the fairness of the experimental test? How effective is the "open-source searching" step in finding the necessary (and up-to-date) code or documentation?

Q4. Alita’s performance across GAIA difficulty levels shows interesting non-dominance: it is outperformed on Level 1 (by OWL/A-World) and Level 3 (by OpenAI DeepResearch). Could the authors provide an analysis of how task difficulty and type influence the effectiveness of Alita's self-evolution mechanism. For instance, is the MCP creation overhead detrimental to solving simple Level 1 tasks, or do the complex requirements of Level 3 tasks simply exceed the current generation capability of the LLM?

Q5. The limitations section correctly highlights the reliance on powerful LLMs for coding. To broaden the practical scope, could the authors provide a more detailed ablation or analysis exploring the impact of using different classes of models (specifically including open-source models like Llama 3 or Qwen 3) for the critical coding and reasoning steps within the MCP creation loop?

---

> ### Author Response · Authors · 2025-12-03
>
> ## Response to W1:
> On one hand, open-sourced coding language model are rapidly imrpoving. Smaller but powerful models like Qwen can already perform well in the mode of Alita. On the other hand, as Claude continues to develop, Alita will be even stronger.
>
> Minimal predefinition is just because the language model can already do deep research on the Internet, find related libraries, and assembly/refine them into MCPs or just use codeact to write MCPs from LLM's internal knowledge. Then there's no need to predefine those tools beforehand.
>
> ## Response to W2 & Q2:
> The cost of MCP creation is not the core issue. This is because the major cost is derived by the web agent browing web pages for multi-turn searching. Compared to the web agent, the MCP creation cost is really low. Also, the search for related libraries spends very little time and low cost in comparison.
>
> ## Response to W3:
> Manually-engineered tools still have an edge currently but as coding language model is rapidly improving and deep research agent is improving, there will be gradually less need for manually-engineered tools.
>
> But actually, for Level 3, the difference purely comes from very complex deep research problems instead of tool use tasks. Our agent is not trained on web searching.
>
>
> ## Response to W4:
> We included the advanced baselines available at the time of wrapping up the paper. Tongyi Deep Research series is around one month after this paper.
>
> ## Response to Q1:
> The 75.15% figure refers to results obtained using Claude-Sonnet-4, while the 72.73% corresponds to results using Claude-3.7-Sonnet; this is explicitly noted in the experimental section.
>
> ## Response to Q2:
> Please refer to response to W2.
>
> ## Response to Q3:
> We don't quite get this question. Other agent also has web agent for access to the Internet. And engineers can read all up-to-date code and documentation to build the agents.
>
> ## Response to Q4:
> The observed non-dominance in specific categories may stem from the fact that other baselines were specifically optimized or for even overfit certain task subsets (e.g., Level 1 tasks). In contrast, Alita treats all tasks uniformly, enabling it to achieve the best overall performance, even if it does not strictly dominate every specific subset.
>
> ## Response to Q5:
> We agree that performance on open-source models is crucial. We conducted a supplementary experiment substituting the closed-source model with Qwen3 8B, achieving a Pass@1 of 61.82% (102/165) on GAIA. This result demonstrates that our framework maintains strong performance even when utilizing open-source models.

---

### Comment · Area_Chair_B1um · 2025-11-28
**please respond to reviewer concerns promptly**

Dear Authors,

A quick reminder that the ICLR 2026 rebuttal window will close in less than a week. To ensure a fair and thorough evaluation, we encourage you to address the reviewers’ concerns as soon as possible.

Thank you for your prompt attention and for helping us keep the process efficient.

AC

---

### Note · Program_Chairs · 2026-01-17
**Submission Desk Rejected by Program Chairs**

The following references in this submission do not refer to real documents and/or have major errors in bibliographic information:

 [13] Agent Team at Ant Group. Aworld: A unified agent playground for computer and phone use tasks, 2025.